# United States Long-Term Trends in Adult BMI (1959–2018): Unraveling the Roots of the Obesity Epidemic

**DOI:** 10.3390/ijerph21010073

**Published:** 2024-01-09

**Authors:** Julia Banas, Acree McDowell Cook, Karina Raygoza-Cortez, Daniel Davila, Melinda L. Irwin, Leah M. Ferrucci, Debbie L. Humphries

**Affiliations:** 1Yale School of Public Health, Yale University, New Haven, CT 06510, USA; juliacbanas@gmail.com (J.B.); elizabeth.mcdowell@yale.edu (A.M.C.); anakarina.raygozacortez@yale.edu (K.R.-C.); daniel.davila@yale.edu (D.D.); melinda.irwin@yale.edu (M.L.I.); leah.ferrucci@yale.edu (L.M.F.); 2Yale Cancer Center, New Haven, CT 06510, USA

**Keywords:** obesity, BMI, socioeconomic disparities, racial disparities, educational disparities, system-thinking approach

## Abstract

The escalating rates of obesity since the 1950s poses a critical public health challenge across all age groups in the United States. While numerous studies have examined cross-sectional disparities across racial, ethnic, and socioeconomic groups, there has been limited research on long-term trends. To address this gap, we analyzed average adult body mass index (BMI) trends from 1959 to 2018, using data from the National Health and Nutrition Examination Survey (NHANES) and the National Health Examination Survey (NHES). Employing time series analysis, we evaluated BMI trends across income, education, and race/ethnicity. The results revealed a consistent upward trajectory in average BMI across all groups over the six-decade period, with no significant differences by income or education levels among high school graduates. However, individuals with less than a high school education displayed a more gradual increase in BMI. Racial disparities were also evident, with Black adults showing higher BMI growth rates compared to White adults, while Hispanic and other racial groups experienced slower increases. These findings underscore the need for systemic interventions to address the ongoing obesity epidemic, emphasizing the importance of research to identify trends over time and a system-thinking approach to inform effective population-level interventions and policy decisions.

## 1. Introduction

Obesity, characterized by a higher-than-normal accumulation of adipose tissue or fat in the body [1], is a pressing public health concern in the United States, with its prevalence steadily rising over the past 20 years [2,3]. By 2016, nearly 40% of American adults were estimated to be obese, and projections suggest this number could rise to almost half of all American adults by 2030 [2]. This trend carries significant implications for public health, as obesity is associated with numerous adverse health effects, including an elevated risk of premature mortality, diabetes, heart disease, cancer, stroke, and various other health complications [4,5].

Body mass index (BMI) is commonly used to assess obesity in large epidemiologic studies. However, its accuracy in reflecting individual obesity can be compromised for various reasons, including variations in muscle mass and the distribution of excess adiposity [6]. An additional limitation is when considering different racial and ethnic groups, as it may be important to use different cut-points for overweight and obesity [6]. Moreover, significant variations in percentage body fat and measurements of visceral adipose tissue (VAT) persist, even when individuals share similar BMI or waist circumference levels across diverse ethnicities [7]. This further underscores the challenge of effectively assessing metabolically healthy versus unhealthy obesity within ethnic groups.

Several factors, encompassing lifestyle choices, socioeconomic variables, genetics, the gut microbiome, the brain-gut axis, physical activity levels, and cultural influences, collectively underscore the complexity of obesity, necessitating a holistic approach [7,8,9,10]. The socioecological model provides a comprehensive lens through which to examine numerous factors contributing to obesity trends. At the societal level, the food environment plays a role, with food insecurity in the United States linked to an increased likelihood of overweight or obesity [11]. Geographical influences have also been seen, with higher obesity rates in specific U.S. states, including Alabama, West Virginia, Louisiana, and Mississippi [12,13].

Data now show an association between childhood and adolescent obesity and persistence into adulthood. Studies show that approximately 55% of obese children remain obese during adolescence, and 80% of obese adolescents will continue to be obese in adulthood. Individuals born between 1956 and 1985 experienced a 20% obesity prevalence in their 20s and 30s, with significant increases observed in boys after 1970 and girls after 1980 [12,14]. Looking at more recent data, obesity can emerge during elementary school years, as evidenced by a 4.5% increase in relative obesity between kindergarten and fifth grade [15]. These data suggest the importance of early intervention and prevention efforts and highlight the need for targeted strategies across different life stages.

Studies of individual-level factors, such as socioeconomic status (SES), race/ethnicity, and physical activity, consistently demonstrate widening disparities. Lower SES groups are particularly affected, with income disparities significantly contributing to these trends [16,17,18,19,20]. The risk factors for obesity vary across racial and ethnic groups, with non-Hispanic Black adults having the highest rates, followed by Mexican Americans, non-Hispanic White Americans, and Asian adults exhibiting the lowest rates [12,21,22]. Despite cross-sectional studies highlighting disparities at specific time points, a longitudinal perspective reveals a commonality in the increasing prevalence of obesity across all racial and ethnic groups [8,23,24]. NHANES data further supports this, indicating a substantial rise in mean BMI for both the highest and lowest income groups between 1960 and 2007, suggesting similar trends even among the most privileged groups [8,25].

This analysis distinguishes itself by offering a longitudinal perspective spanning several decades, aiming to examine population-level obesity trends and prevalence across diverse racial and ethnic groups. The primary objective is to characterize trends in average BMI over time by individual-level factors such as socioeconomic status (SES), race/ethnicity, and education and to assess differences in time trends across SES, race/ethnicity, and education groups. Such an analysis can contribute important insights into the long-term patterns of obesity. Building upon previous longitudinal analyses [25] and analyzing the National Health and Nutrition Examination Survey (NHANES) and the National Health Examination Survey (NHES) data from 1959 to 2020, our hypothesis posited that trends in BMI in the United States would not exhibit statistically significant differences across income, race, ethnicity, and education groups.

## 2. Materials and Methods

### 2.1. Data Collection

The dataset for this study integrates information from multiple waves of NHANESs (1971–2020) and the first wave of the NHESs (1959–1962). NHANES data are collected biannually, with approximately 5000 adults per wave. In the NHANES, sociodemographic data, medical history, and health measurements are collected through interviews and physical health examinations.

### 2.2. Data Analysis

SAS 9.4 was employed for data analysis. Participants were restricted to ages 20 and above. BMI was used as a continuous variable.

Household income standardization was performed using data from the Current Population Survey (CPS) and then categorized into deciles. The lowest decile was designated as low income, and the highest decile was assigned to indicate high income.

For race, the NHES, NHANES I, and NHANES II collected participants’ races as White, Black, or other. The NHANES III added Mexican American to this category, and other Hispanic was added starting in 1999. In 2011, the category began to include non-Hispanic Asian.

In the NHES, education data included categories such as 1 to 4 years of school, 5 to 8 years, 9 to 12 years, 1 to 2 years of college, 3 to 4 years of college, and over 4 years of college. For the NHANES, education was classified into Less Than 9th Grade, 9–11th grade, High School Grad/GED or Equivalent, Some College or AA degree, and College Graduate or above. Since the NHANES did not collect data solely on a high school diploma, 1 to 8 years of school was coded as less than high school education, 9 to 12 years as high school education, and 1 to 2 years of college and above as greater than high school education.

Time Series (ITS) analysis was employed to analyze obesity trends across income, education, and racial/ethnic groups. The analysis involved fitting regression lines for each group, utilizing the midpoint year as the time variable. Slopes were compared using interaction variables for year and income, year and education, or year and racial/ethnic group. F-tests and interaction terms in regression models were used to compare regression slopes. We also explored interactions among the three variables of income, education, and race/ethnicity. Mean BMI by year and category was calculated and graphed using OriginPro 2021.

## 3. Results

### 3.1. Study Population

The total sample size including all waves was 46,956 individuals. The mean BMI of each income, education, or racial/ethnic group by survey wave can be found in Table A1.

### 3.2. Income

Mean BMI for the top and bottom income deciles, while consistently higher for the bottom income decile, were not significantly different in their slopes (Figure 1; *p*-value 0.28). Regression analysis demonstrated a statistically significant positive trend over the years (estimate: 0.089, *p*-value < 0.001), indicating an overall increase in BMI (Table 1).

The interaction between year and income was not significant, indicating that changes in BMI over time were not significantly different between the two income groups (*p*-value = 0.25).

Figure A1 displays the regression lines for the top income decile and bottom income decile, where the slopes were not significantly different from each other.

### 3.3. Education

Adults with less than a high school education have a significantly different trend in BMI growth from individuals with more than a high school education (Table 1). While the trends in BMI for adults with a high school education and more than a high school education have been similar over time, since the mid-1980s adults with less than high school education have a slower increase in BMI (Figure 2). The difference in slopes of those with a high school diploma and those with higher education was not significantly different (Table 1 and Figure A2).

### 3.4. Race/Ethnicity

Black, Hispanic and adults who self-report other as their race/ethnicity had significantly different BMI growth trends than their White counterparts (Figure 3; *p*-values = 0.002, <0.001 and <0.001, respectively). The interaction terms with time for Black, Hispanic, and other race and ethnicity are statistically significant, signifying diverse trends in BMI over time for each racial and ethnic group (Table 1 and Figure A3).

### 3.5. Interactions

#### 3.5.1. Income and Education

Trends in average BMI for adults from the lowest income decile with less than high school education were not significantly different from those in the highest income decile with less than a high school education (Table A2). Those in the lowest income decile with a high school diploma or more than a high school education had significant differences in BMI trends, as did those in the highest income decile with a high school diploma (Table A2). Figure A4 shows the fitted regression lines.

#### 3.5.2. Income and Race/Ethnicity

Trends in average BMI among Black, Hispanic, and White adults in the lowest income decile were not significantly different from White adults in the highest income decile. However, adults of other races in the highest income decile and lowest income decile had significantly different slopes compared to White adults in the highest income decile (Table A3). Figure A4 shows the fitted regression lines.

#### 3.5.3. Education and Race/Ethnicity

When breaking down BMI trends by education and race/ethnicity, Hispanic adults at all levels of education did not have significantly different BMI trends from White adults with less than high school education. Black adults with less than high school education also did not have significantly different slopes compared to the White adults with less than high school education, in addition to adults in the other race category who had more than high school education. All other groups had significantly different slopes compared to White adults with less than high school education (Table A4). The fitted regression lines are shown in Figure A5.

## 4. Discussion

These findings carry significant implications for addressing obesity in the United States as we found that disparities in BMI trends are not growing or narrowing based on groups of income or education. Further examination of interactions between income, education, and race/ethnicity reveal that subgroups are experiencing similar BMI trends. Notably, both the highest and lowest income deciles exhibit comparable increases in BMI over time, emphasizing the need for broader public health strategies despite disparities in BMI across income levels. While racial/ethnic groups had significant differences in BMI trends, the results among education groups were mixed. Individuals with a high school degree and those with higher education experienced a similar increase in BMI over time, while those with less than high school education experienced a slower increase in BMI over time.

The implications of several trendlines not differing are significant for addressing obesity in the United States. With all fitted regression lines displaying positive slopes, the trajectory of this public health problem is expected to become increasingly prevalent over time. The results pertaining to income are particularly striking, revealing that despite existing disparities in BMI across income levels, both the top and bottom income groups in this dataset are on the same upward trajectory. While income disparities introduce economic considerations influencing access to health-related resources [26], the lack of a widening or narrowing gap suggests a persistent issue that necessitates comprehensive societal changes to address. The gap is not widening, but it is also not narrowing, which shows that there is much to be done to address this problem and a widespread, societal change may be necessary. This need for broader public health strategies is echoed in the results for education groups, where those with a high school degree and those with higher education both experienced the same increase in BMI over time. This is another disparity present that does not appear to be changing. However, those with less than a high school diploma did experience a significantly different trendline than the other education groups, presenting a noteworthy departure from the anticipated impact of higher education on adopting healthier lifestyles, as suggested by previous research [27]. These findings are similar to those of Ljungvall and Zimmerman [25], who find time trends in obesity to be similar among income, educational, and racial/ethnic groups overall, with some exceptions. These results also show that obesity is increasing among every group, which is consistent with Wang et al.’s [2] and Zimmerman et al.’s [8] projections.

This study distinguishes itself through several strengths, primarily by prioritizing long-term trend data and utilizing a nationally representative dataset—an approach that sets it apart from the existing literature, which frequently relies on cross-sectional and short-term estimates of BMI. Furthermore, the incorporation of time series analysis methods enhances the depth of our analytical approach by allowing us to examine patterns and trends over an extended period, capturing the temporal dynamics of the relationships between income, education, race/ethnicity, and average BMI. Unlike cross-sectional studies that provide a snapshot at a specific point in time, time series analysis enables the exploration of how variables change over time.

However, this study has limitations including restricted data on race/ethnicity stemming from categorization changes in the NHANES and the use of BMI as an outcome rather than a gold standard measure of body composition. Moreover, it is essential to consider additional factors in the BMI analysis by group. The analysis employed the White population as the control group, potentially overlooking other between-group variations that could influence various outcomes.

### Future Research Directions

Future longitudinal analyses are needed that consider obesity rates and the synergistic effects of many other variables that could be relevant, including food marketing, antibiotic use, sleep duration, screen time, the gut microbiome, stress, and chemical exposures [7,8,28,29]. For example, examining the relationship between stress and obesity, along with the higher cost of healthy diets [30,31,32]. These potential consequences should be considered in addressing all aspects of the obesity epidemic due to the well-established link between increasing BMI and higher all-cause mortality [33].

As depicted in Figure 4, the socioecological model serves as a guiding framework for presenting the multifaceted drivers of the obesity epidemic. Researchers should focus on unravelling the dynamic interactions among variables at different levels, considering the temporal dynamics highlighted by time series analysis methods. The socioecological perspective prompts exploration of the interconnectedness of various determinants, such as screen time, sleep, stress, and diet, as highlighted by previous research [7,8,28,29,34,35,36]. More of the drivers of the obesity epidemic may need to be understood to reduce the epidemic at a society level, while simultaneously moving forward with evidence-based interventions at the individual level. There is evidence for several interrelated disease physiologies and causes [35,36]. For example, screen use and sleep are related to each other and to obesity [37]. The interrelated nature of diet, sleep, stress, and the microbiome, and their relationships to the development of obesity, underscores the need for systematic reviews and likely more original research.

Our study strongly advocates for comprehensive structural changes at all societal levels to effectively prevent obesity. Our study shows that current interventions, primarily focused on specific subgroups categorized by income, education, or race/ethnicity, have demonstrated limited effectiveness in mitigating disparities and decreasing the overall prevalence of obesity. To find effective solutions, we must also examine the unintended consequences of the interventions in place, for example, in the context of the problematic marketing of unhealthy food [8], as market interventions, if done incorrectly, could have devastating economic effects on both the marketing and food product industries.

## 5. Conclusions

Our analysis reveals that obesity rates are consistently on the rise across all income levels, racial/ethnic backgrounds, and educational attainments. This increase challenges conventional expectations, emphasizing the need for comprehensive, population-wide interventions. Regardless of socioeconomic factors, to combat the obesity epidemic successfully, we must shift our focus from subgroup-specific strategies to comprehensive reforms that encompass the food industry, infrastructure, food marketing and resource allocation within communities.

## Figures and Tables

**Figure 1 ijerph-21-00073-f001:**
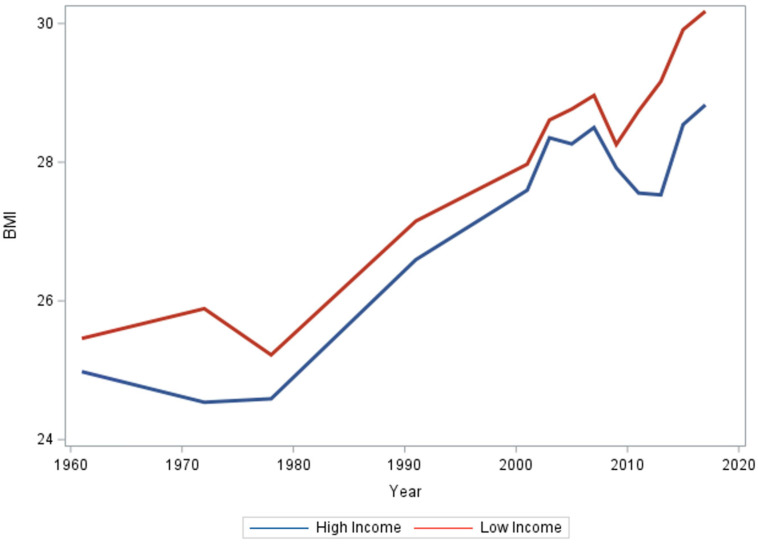
Mean BMI of the top income decile and the bottom income decile from 1961 to 2017.

**Figure 2 ijerph-21-00073-f002:**
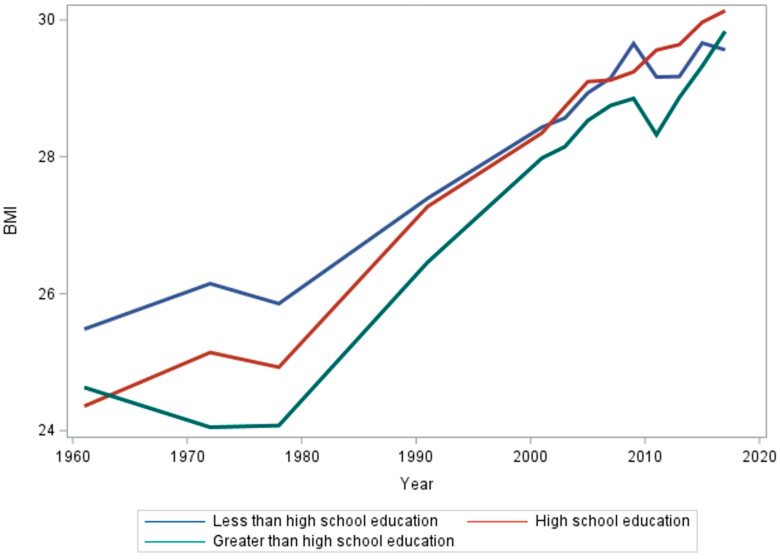
Mean BMI of each education group from 1961 to 2017.

**Figure 3 ijerph-21-00073-f003:**
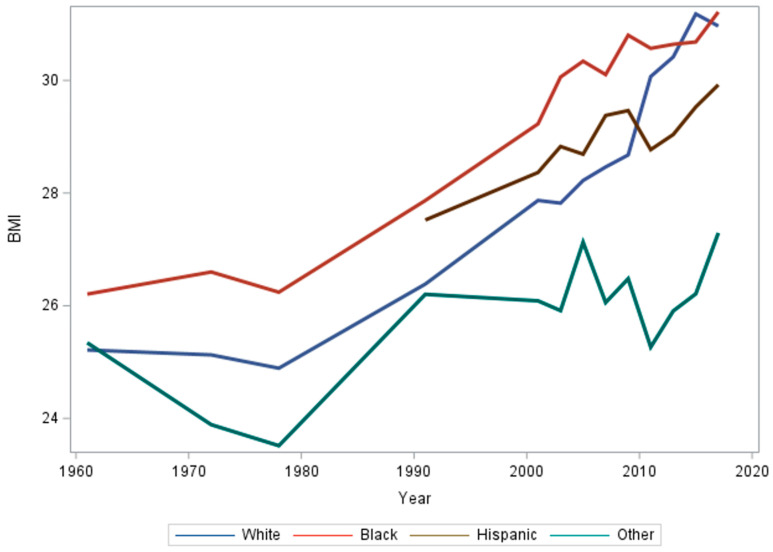
Mean BMI of each racial/ethnic group from 1961 to 2017. Note: data on race/ethnicity was collected differently throughout NHES and NHANES history. In this figure, “Other” includes Mexican American and other Hispanic until NHANES III, where these categories were included in the survey. Starting in 1991, this analysis includes a group for Hispanic participants, and “Other” refers to races/ethnicities that are not Black, White, or Hispanic.

**Figure 4 ijerph-21-00073-f004:**
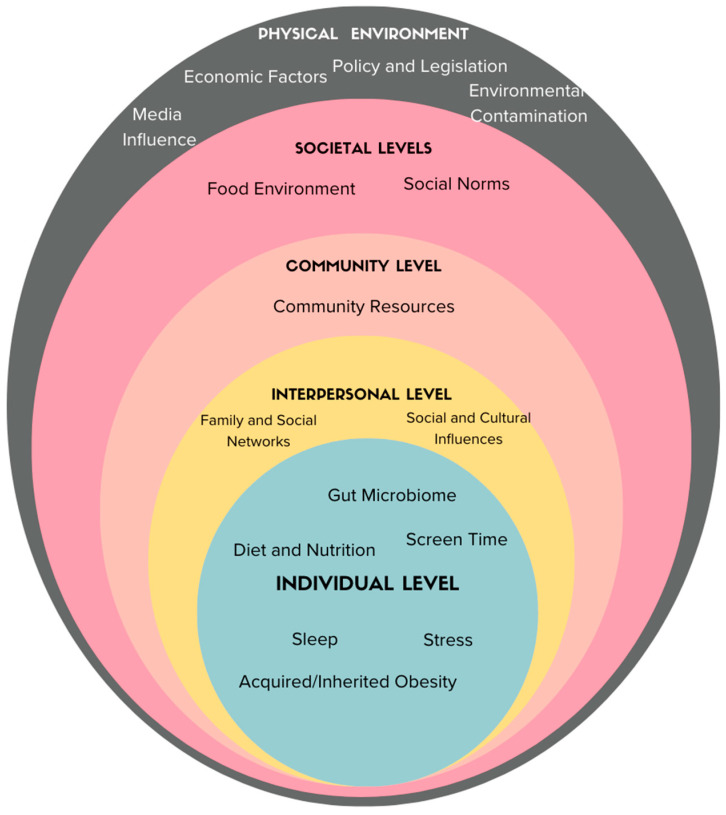
Drivers of obesity presented in the framework of the SocioEcological Model. This figure delineates the variety of potential influences and drivers of the obesity epidemic, as identified in the literature, oriented within the socioecological model to illustrate the diverse levels at which specific causes exert influence.

**Table 1 ijerph-21-00073-t001:** Time Series analysis of longitudinal trends in BMI (1959–2018) by income, education and race/ethnicity.

Variable	Estimate	Standard Error	*p*-Value
Income			
Year (changes over time)	0.089	0.004	<0.001
Low income (reference)			
High income	10.938	10.200	0.284
Interaction	−0.006	0.005	0.253
Education			
Year	0.124	0.002	<0.001
Less than high school	84.951	5.521	<0.001
High school diploma	7.146	6.486	0.271
Greater than high school (reference)			
Interaction (less than high school)	−0.042	0.003	<0.001
Interaction (high school diploma)	−0.003	0.003	0.318
Race/ethnicity			
Year	0.0956	0.002	<0.001
White (reference)			
Black	−19.424	6.342	0.002
Hispanic	33.439	9.538	0.001
Other	114.610	12.528	<0.001
Interaction (BBlack)	0.010	0.003	0.001
Interaction (Hispanic)	−0.016	0.005	0.006
Interaction (other)	−0.058	0.006	<0.001

## Data Availability

The data supporting the reported results in this study are publicly available and can be accessed through the National Health and Nutrition Examination Survey (NHANES) website at [https://www.cdc.gov/nchs/nhanes/index.htm accessed on 15 November 2020]. As the data used in this research are already in the public domain, no additional datasets were generated during the study. All relevant information for replicating the analysis is provided in the manuscript.

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
