# Peer review of "United States Long-Term Trends in Adult BMI (1959–2018): Unraveling the Roots of the Obesity Epidemic"

_ijerph, 2024, doi:10.3390/ijerph21010073_

Round 1

Reviewer 1 Report

Comments and Suggestions for Authors

Dear Authors,

Congratulations on undertaking such extensive research. However, in my opinion, the manuscript requires refinement.

Comments:

1. Please clearly state the purpose of the research and the hypothesis.

2. Please verify the keywords: what is the meaning of entering obesity trends, obesity causes? Isn't "obesity" enough? Does the keyword systematic interventions make sense?

3. Tables: different numbers of digits after the dot (e.g. 10.9386 ; -0.00584). Please standardize this.

4. Fig. 4 - what data was it based on? What methodology was adopted?

 5. Conclusions: please clearly define the purpose of the research and then develop conclusions relating in detail to your research. The current conclusions could have been developed without analysis of scientific research. These are very general terms.

Author Response

  1. Please clearly state the purpose of the research and the hypothesis.

We appreciate the reviewer’s encouragement to further clarify the purpose and hypothesis.  We have done so in the introduction with the following language (line numbers 72-78):

“The primary objective is to address disparities in individual-level factors such as socio-economic status (SES), race/ethnicity, and education to provide insights into the patterns of obesity. Building upon previous longitudinal analyses[25] and analyzing National Health and Nutrition Examination Survey (NHANES) and National Health Examination Survey (NHES) data from 1959 to 2020, our hypothesis posited that trends in BMI in the United States would not exhibit statistically significant differences across income, race, ethnicity, and education groups.”

  1. Please verify the keywords: what is the meaning of entering obesity trends, obesity causes? Isn't "obesity" enough? Does the keyword systematic interventions make sense?

Thank you for your feedback. We have revised our keywords accordingly.

  1. Tables: different numbers of digits after the dot (e.g. 10.9386 ; -0.00584). Please standardize this.

We appreciate your attention to detail. We have addressed the inconsistency in decimal placement for better standardization in our tables.

  1. Fig. 4 - what data was it based on? What methodology was adopted?

We agree this is an important question.  The figure builds on the visual of the socioecological model.  After reviewing the literature on causes of obesity we chose to present potential influences on and drivers of obesity within the framing of the socioecological model as a way to emphasize the diverse levels at which specific causes exert influence.  References are highlighted in the text (lines 247-248).  In addition, we revised the figure caption as follows: (lines 256-258)

Figure 4. Drivers of Obesity presented in the framework of the Socio-Ecological Model. This figure delineates the variety of potential influences on and drivers of the obesity epidemic, as identified in the literature, oriented within the socioecological model to illustrate the diverse levels at which specific causes exert influence.

  1. Conclusions: please clearly define the purpose of the research and then develop conclusions relating in detail to your research. The current conclusions could have been developed without analysis of scientific research. These are very general terms.

We acknowledge the need for clearer articulation of the research purpose and more detailed conclusions. We have refined our conclusions to better align with our study specific aims.  See additional text at lines 264 to 272.

Our analysis reveals that obesity rates are consistently on the rise across all income levels, racial/ethnic backgrounds, and educational attainments, with disparities in av-erage BMI that existed in 1960 by income and most race/ethnicity groups remaining sixty years later. The patterns of increase, with disparities unchanged sixty years later, chal-lenge conventional expectations, emphasizing the need for comprehensive, popula-tion-wide interventions. Regardless of socioeconomic factors, to combat the obesity epidemic successfully, we must shift our focus from subgroup-specific strategies to comprehensive reforms that encompass the food industry, infrastructure, food mar-keting and resource allocation within communities.

Reviewer 2 Report

Comments and Suggestions for Authors

Overall: The data from this study is informative and revealing of both the significant of the BMI trends over time and whether they diverge across income, ethnicity and education. I am concerned that a lot of the focus of the manuscript extends into broader assumptions of what may be prompting these changes. This strays from the data and what is known from the data. For instance, while the social ecological model certainly plays into the broader challenges that make up BMI trends, the researchers have only focused on differences across income and ethnicity and that is really where the introduction and discussion should should align. The data itself is unable to state whether this is due to food insecurity or other factors in the social ecological model presented. Therefore this should really move into the future longitudinal analyses section where future research should consider how the varying components mentioned in the model may play into preventative efforts. Therefore I would suggest the model and lines 189 to 200 move into future research surrounding preventative efforts or be removed as they extend beyond the data itself. I would make this comment more specific to why ethnicity, income and education appear to be telling a different story with BMI trends more so that broad potential for preventative impact.

Good writing and references. Easy to read and I like the figures. Only specific comment is below:

Consider adding in a sentence and reference at line 38 about the challenge of assessing metabolically healthy vs unhealthy obesity using BMI and how these differ in ethnic groups (see Lear).

Author Response

  1. Overall: The data from this study is informative and revealing of both the significant of the BMI trends over time and whether they diverge across income, ethnicity and education. 

We appreciate the reviewer’s comments.

  1. I am concerned that a lot of the focus of the manuscript extends into broader assumptions of what may be prompting these changes. This strays from the data and what is known from the data. For instance, while the social ecological model certainly plays into the broader challenges that make up BMI trends, the researchers have only focused on differences across income and ethnicity and that is really where the introduction and discussion should should align. The data itself is unable to state whether this is due to food insecurity or other factors in the social ecological model presented. Therefore this should really move into the future longitudinal analyses section where future research should consider how the varying components mentioned in the model may play into preventative efforts. Therefore, I would suggest the model and lines 189 to 200 move into future research surrounding preventative efforts or be removed as they extend beyond the data itself.

Thank you for your thoughtful feedback. We appreciate your concern regarding the broader assumptions in the manuscript. In response to your comment, we have made revisions to better align the introduction and discussion with the data. We have added explicit language highlighting our results [lines 191-211] and have repositioned the discussion of the social ecological model to future research directions for clarity. We have separated the relevant parts of the discussion into future longitudinal analyses as recommended and added additional details. [see lines 230-257]

  1. I would make this comment more specific to why ethnicity, income and education appear to be telling a different story with BMI trends more so that broad potential for preventative impact.

We have revised the relevant section of the discussion to provide a more detailed exploration of the unique factors influencing BMI trends within each demographic category. The updated paragraph now explicitly looks into the role of cultural influences, economic considerations, and educational backgrounds in shaping individuals' BMI trajectories (Lines 189-210). Additionally, we have incorporated additional references to support the expanded discussion on these unique factors.

These findings carry significant implications for addressing obesity in the United States as we found that disparities in BMI trends are remaining constant based on groups of income and education. Further examination of interactions between income, education, and race/ reveal that subgroups are experiencing similar BMI trends. Notably, both the highest and lowest income deciles exhibit comparable increases in BMI over time, emphasizing the need for broader public health strategies despite disparities in average BMI across income levels. While racial/ethnic groups had significant differences in BMI trends, the results among education groups were mixed. Individuals with a high school degree and those with higher education experienced a similar increase in BMI over time, while those with less than high school education experienced a slower increase in BMI over time.

The implications of several trendlines not differing are significant for addressing obesity in the United States. With all fitted regression lines displaying positive slopes, the prevalence of obesity is expected to continue to increase in the near future. The results pertaining to income are particularly striking, revealing that despite existing disparities in BMI across income levels, both the top and bottom income groups in this dataset are on the same upward trajectory. While income disparities introduce economic considerations influencing access to health-related resources, [26] the lack of a widening or narrowing gap suggests a persistent issue that necessitates comprehensive societal changes to address. This need for broader public health strategies is echoed in the results for education groups, where those with a high school degree and those with higher education experienced similar increases in BMI over time and the disparities between the two groups are not changing. However, those with less than a high school diploma did experience a significantly different trendline, with a slower increase in average BMI than the other education groups, presenting a noteworthy departure from the anticipated impact of higher education contributing to increased adoption of healthier lifestyles, as suggested by previous research. [27] These findings are similar to those of Ljungvall and Zimmerman[25], who find time trends in obesity to be similar among income, educational, and racial/ethnic groups overall, with some exceptions. These results also show that obesity is increasing among every group, which is consistent with Wang et al.’s[2] projections.

  1. Good writing and references. Easy to read and I like the figures. 

Thanks for this feedback!

  1. Only specific comment is below:  Consider adding in a sentence and reference at line 38 about the challenge of assessing metabolically healthy vs unhealthy obesity using BMI and how these differ in ethnic groups (see Lear).

We have added additional materials as recommended. See lines 36-44.

Body mass index (BMI) is commonly used to assess obesity in large epidemiologic studies. However its accuracy in reflecting individual obesity can be compromised for various reasons, including variations in muscle mass and the distribution of excess adiposity. [6] An additional limitation is when considering different racial and ethnic groups, as it may be important to use different cut-points for overweight and obesity.[6] Moreover, significant variations in percentage body fat and measurements of visceral adipose tissue (VAT) persist, even when individuals share similar BMI or waist circumference levels across diverse ethnicities. [7] This further underscores the challenge of effectively assessing metabolically healthy versus unhealthy obesity within ethnic groups.